materials science/chemical engineering

water-soluble polymer, cement slurry, constant rheology

**Authors for correspondence:**
Huajie Liu
e-mail: liuhuajieupc@163.com
Jiapei Du
e-mail: jiapei.du@student.rmit.edu.au

This article has been edited by the Royal Society of Chemistry, including the commissioning, peer review process and editorial aspects up to the point of acceptance.

# A novel hydrophobically associating water-soluble polymer used as constant rheology agent for cement slurry

Yuhuan Bu[1,2], Mengran Xu[1,2], Huajie Liu[1,2],
Annan Zhou[3], Jiapei Du[3], Xin Yang[4] and Shenglai Guo[1,2]

[1]Key Laboratory of Unconventional Oil & Gas Development (China University of Petroleum (East China)), Ministry of Education, Qingdao 266580, People's Republic of China
[2]School of Petroleum Engineering, China University of Petroleum (East China), Qingdao 266580, People's Republic of China
[3]School of Engineering, Royal Melbourne Institute of Technology, Victoria 3001, Australia
[4]CPOE Boxing Engineering Technology Co. Ltd., Tianjin 300450, People's Republic of China

HL, 0000-0001-8754-7145; JD, 0000-0002-4309-9528;
SG, 0000-0002-5881-4942

During the process of well cementing in deep water, the cement slurry experiences a wide range of temperature variation from low temperature at seabed to high temperature in downhole. The elevated temperature affects the rheology of cement slurry. The change of rheology of cement slurry could influence the safety of cementing operation. The aim of this paper is to develop a new kind of hydrophobically associating water-soluble polymer (NHAWP) as an additive to prepare a constant rheology oil well cement slurry, which can be used at temperature range from 4°C to 90°C. The acrylamide, 2-acrylamide-2-methylpropionic acid and stearyl methylacrylate were applied to synthesize the NHAWP by the inverse microemulsion polymerization. Test results indicate that the critical association temperature of NHAWP is 45°C. The critical association temperature is independent of NHAWP concentration, salt concentration and alkalinity of solution. When the temperature is below 45°C, NHAWP shows little influence on the viscosity of solution. When the temperature is above 45°C, the NHAWP forms spatial network structure by intermolecular hydrophobic association and thus increases the viscosity of solution significantly. The NHAWP also displays good thermal stability and excellent salt and alkali resistance properties. In addition, the NHAWP shows nearly no negative influence on the basic properties of cement slurry, which indicates that the NHAWP can be used

as a constant rheology agent to prepare a cement slurry with constant rheology in the temperature range of 4°C to 90°C.

## 1. Introduction

Rheology is one of the most important characteristics of oil well cement slurry [1,2]. During the cementing operation of oil and gas well, cement slurry is injected into the annulus between the casing and formation through the casing strings [3,4]. In the meantime, the drilling fluid, which presents in the narrow annulus, will be displaced by cement slurry. Therefore, controlling the viscosity of cement slurries within a proper range is vital for the successful displacement of drilling fluid [5]. If the viscosity of cement slurry is high, a large pump pressure is required to achieve a good displacement efficiency. However, the large pump pressure leads to formation breakdown, thereby results in well-cementing accidents [6]. If a low pump pressure is used, the displacement efficiency will be poor and the cementing quality will be weak [7,8]. If the viscosity of the cement slurry is reduced to enhance the displacement efficiency, the settlement stability of the cement slurry becomes poor, and therefore free water will come out from the slurry and accumulate in the upper layer of cement [9]. After that, cross-flow channels, which allow oil, gas and water flow through, will be formed and result in well-cementing failure. Therefore, in order to achieve the best displacement efficiency and ensure cementing safety, it is necessary to optimize the rheology of cementing slurry before cementing operation.

However, the rheological property of cement slurry usually changes with temperature variations [10,11]. When temperature increases, the viscosity of cement slurry decreases [12]. Generally, the temperature of downhole increases gradually with the increase of well depth (the geothermal gradient is 3°C/100 m). Therefore, the viscosity of cement slurry varies with the changing of well depth, which makes the rheology design of cement slurry very difficult. In particular, during the process of offshore deep-water cementing, because the temperature of deep-water mudline usually ranges from 2°C to 4°C [13], the cement slurry will undergo a wide temperature range from mudline to downhole. For the shallow formation, due to low temperature, the viscosity of the cement slurry is high, which leads to the difficulty of injecting the cement slurry. Therefore, low-viscosity cement slurry is a benefit for pumping in shallow formation of deep-water. In the meantime, when the low-viscosity cement slurry reaches the downhole, the cement slurry would be diluted seriously due to the high temperature [10]. This phenomenon causes the weak settlement stability of cement slurry and therefore affects the quality of well cementing.

Therefore, if the cement slurry possesses constant rheology at different temperatures, the quality of well cementing will be improved and the safety of cementing operation will be ensured. The application of rheology-control additives is the main method to adjust the rheology of cement slurry [14]. An effective rheology-control additive should increase the viscosity of cement slurry at high temperature, while it should not show much influence on the rheology of cement slurry at low temperature. Previous studies showed that hydrophobically associating water-soluble polymers (HAWP) show a thermal tackifying property [15,16]. Muñoz-López *et al.* [17] reported an associating multi-block copolymer electrolytes mediated by radical addition-fragmentation chain transfer technique. The influence of hydrophobic length was illustrated to play an essential role in the rheological properties of copolymers. Jiménez-Regalado *et al.* [18] investigated the phase behaviour of hydrophobically modified polyacrylamides. Results show that viscosity of the polymer system is depressed due to a local segregation between the two copolymers. Chen *et al.* [19] proposed a new kind of HAWP, which can be used as tackifier for cement slurry at high temperature. However, the cement slurry with this kind of HAWP also displays a pretty high viscosity at low temperature. In this paper, a new kind of HAWP, named NHAWP, was developed to make the cement slurry maintain constant rheology. First, the molecular structure of NHAWP was designed. After that, the viscosity–temperature characteristics, salt resistance and alkali resistance properties of NHAWP were characterized and evaluated. The mechanism of NHAWP affected on the viscosity of cement slurry at different temperatures was revealed. Finally, the influence of NHAWP on the properties of cement slurry was evaluated.

## 2. Material and methods

### 2.1. Materials

Acrylamide (AM), 2-acrylamide-2-methylpropionic acid (AMPS), stearyl methylacrylate (SMA), sodium hydroxide (NaOH), sodium bisulfite and ammonium persulfate were obtained from Sinopharm

**Table 1.** Chemical composition of class G oil well cement.

| chemical | CaO | SiO$_2$ | Fe$_2$O$_3$ | Al$_2$O$_3$ | SO$_3$ | MgO | K$_2$O | loss on ignition |
|---|---|---|---|---|---|---|---|---|
| component (wt%) | 65.60 | 22.70 | 4.81 | 3.39 | 1.21 | 0.90 | 0.37 | 0.49 |

Chemical Reagent Co., Shanghai, China. Reactive emulsifier (ER-20) was provided by Nanjing Qinghai Trading Co., Ltd. The class G oil well cement (600 mesh) used in this study, according to American Petroleum Institute (API) 10B-2-2013 and ISO 10426-1, was obtained from Jiahua Special Cement Limited Company, Sichuan province, China. The chemical composition of class G oil well cement is shown in table 1. Fluid loss additive (polycarboxylic acids), retarder (polycarboxylic acids), dispersant (sulfonated aldehydes and ketones) and defoamer were obtained from BO-XING engineering science and technology company of CNPC, Tianjin, China. A basic cement slurry system, which is commonly used for deep-water well cementing, was selected to investigate the effect of NHAWP on cement slurry. The basic formula of the cement slurry is as follows: class G oil well cement + 0.8% dispersant + 5% fluid loss additive + 0.5% defoamer + 0.1% retarder + 44% water. It should be noted that the percentage of additives and water were all calculated by the weight of cement.

## 2.2. Methods

### 2.2.1. Molecular structure design of new kind of hydrophobically associating water-soluble polymer

The HAWP is a kind of water-soluble polymer with a small number of hydrophobic groups in the hydrophilic macromolecular chain [20]. In cement paste, the hydrophobic groups of HAWP can hold together due to hydrophobic interaction, resulting in intramolecular and intermolecular association of macromolecular chains, as shown in figure 1. The intermolecular association can form a supramolecular structure, which is a dynamically physical cross-linked network. This structure could increase the viscosity of solution obviously [20]. Because high temperature enhances the solvent polarity, which can strengthen the hydrophobic association, HAWP shows thermal tackifying effect.

Therefore, the NHAWP should be a kind of HAWP, which can be used to adjust the rheology of cement slurry at high temperature. In the meantime, to achieve a constant rheology property, the NHAWP is not expected to increase the viscosity of cement slurry at low temperature. To achieve this purpose, the molecular chain of NHAWP should be hardly stretching at low temperature, but easily extends at high temperature in solution. Because the stretching degree of the molecular chain relates to the solubility of polymer, it is necessary to adjust the dissolution characteristics of NHAWP in cement paste. In addition, the high salinity and alkaline environment of cement paste may have some effect on the hydrophobic association of HAWP [21]. Therefore, when designing the molecular structure of NHAWP, the dissolution characteristics of NHAWP, the high salinity and alkaline environment of cement slurry, should also be considered.

The solubility of HAWP relates to the performance of hydrophobic groups. In general, the stronger the hydrophobicity of hydrophobic groups (e.g. increase the length of carbon chain), the weaker the solubility of polymers [22]. SMA is a comb-structure monomer, as well as an alkyl acrylate with long side-chains of 18 carbon atoms (figure 2a). Because hydrophobic group chain and C=C bond are connected by ester group and the hydrophobic group chain is long enough, the SMA shows good hydrophobicity [23]. Furthermore, the unstable C=C bond make the SMA copolymerizing with other unsaturated monomers easily. Therefore, SMA is selected as the hydrophobic monomer for synthesizing the NHAWP. The AMPS possesses a strong anion group ($-SO_3^-$), which can repel OH$^-$. Meanwhile, it is insensitive to the attack of cations (figure 2b). Because there are two methyl substituents in the molecular chain, AMPS shows the steric effect and thereby obtains the ability to prevent the attack of OH$^-$ and cation. Therefore, AMPS, which displays good salt and alkali resistance, is selected as hydrophilic monomer for synthesizing the NHAWP. AM, which contains unstable C=C bond, could perform polymerization under the action of radicals or non-ionic initiators (figure 2c). In the meantime, the C-C bond in AM can act as the connecting bond, and the polymers synthesized by AM show high thermal stability [24]. Therefore, AM is selected as the monomer for synthesizing the main chain to ensure the thermal stability of NHAWP. The final molecular structure of developed NHAWP is shown in figure 2d. Moreover, the emulsifier (ER-20) can also act as a hydrophobic monomer, which can polymerize at the same time as SMA due to it possessing the C=C bond.

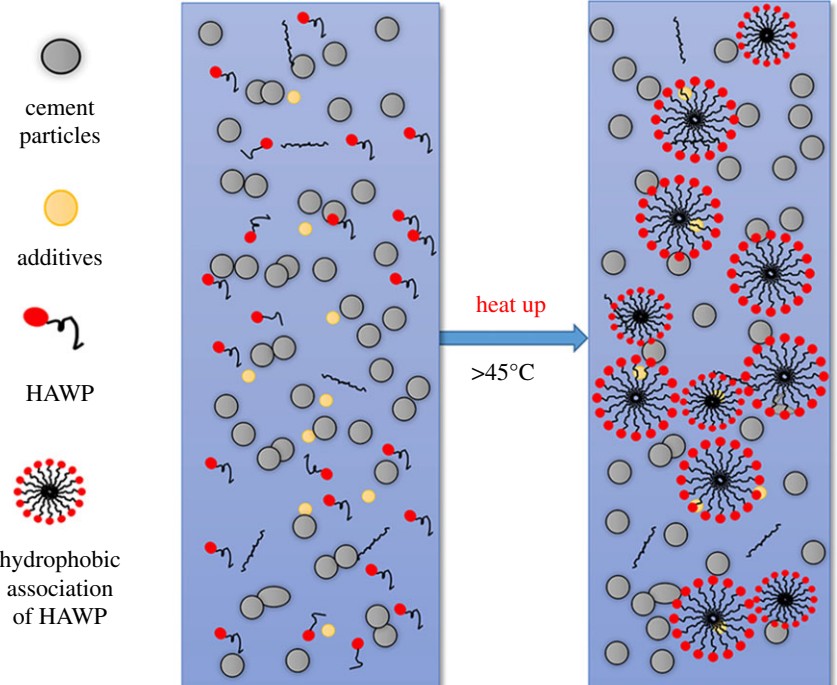

**Figure 1.** Thermal tackifying mechanism illustration HAWP.

**Figure 2.** Molecular structure of monomers for synthesizing the NHAWP. (*a*) SMA; (*b*) AMPS; (*c*) AM; (*d*) molecular formula of the developed NHAWP.

## 2.2.2. Synthesis of new kind of hydrophobically associating water-soluble polymer

The NHAWP was synthesized by the inverse microemulsion polymerization method [25]. The synthesis procedure is shown in figure 3 and the detailed procedures are as follows: first, 100 g distilled water was added into a beaker. After that, AM and AMPS with the mass ratio of 3 : 1 (i.e. mole ratio of 9 : 1) were added to the beaker. The magnetic stirrer was used to make AM and AMPS completely dissolved to form solution-I. Then, 20% of NaOH solution was added into solution-I to adjust the pH of solution-I to a neutral state. Subsequently, the solution-I was heated with a stirring rate of 1000 r.p.m. When the temperature of solution-I raised to 50°C, non-ionic reactive emulsifier and SMA were added into solution-I. The dosage of non-ionic reactive emulsifier and SMA were 21.5% and 25% by the total

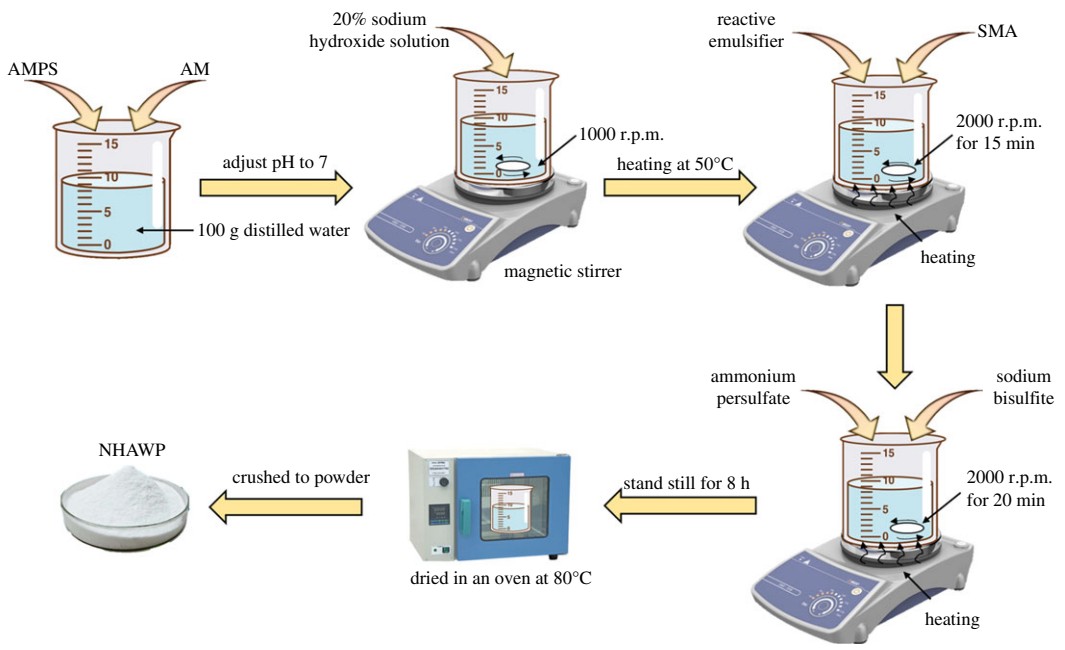

**Figure 3.** The synthesis procedure of NHAWP.

mass of AM and AMPS, respectively. It shall be noted that the mass ratio of hydrophobic monomer and hydrophilic monomer has been pre-adjusted. The effect of hydrophobic monomer on rheological property of cement was also studied in §3.1. Then, the stirring speed was increased to 2000 r.p.m. to emulsify the mixture. After stirring for 15 min, 0.5% dosage of initiator whose mass ratio of ammonium persulfate to sodium bisulfite was 1 : 1 was dripped into the mixture. After that, continue stirring the mixture for 20 min and then keep the mixture stand still for 8 h. The product was dried in an oven at 80°C and crushed to powder. The Mn and Mw of NHAWP are 6259 and 6727, respectively. This result shows that the NHAWP is a narrow molecular weight distribution polymer (Mw : Mn = 1.065) with stable performance.

### 2.2.3. Characterization of new kind of hydrophobically associating water-soluble polymer

#### 2.2.3.1. Molecular structure

The NHAWP was synthesized by copolymerization of aqueous monomers (AM, AMPS), hydrophobic monomer (SMA) and emulsifier (ER-20). In order to verify that NHAWP is a product synthesized by copolymerization of these four monomers, Fourier transform infrared spectroscopy (FT-IR) was used to analyse the molecular structure of NHAWP. Potassium bromide crystal was used to dilute the sample. The wavelength range was 4000–400 cm$^{-1}$.

#### 2.2.3.2. Thermal stability

TG 209F3-type thermogravimetric analyser was used to analyse the thermal stability of NHAWP. The initial mass of the sample was 2.75 mg, and the testing temperature ranges from 30°C to 600°C. The heating rate was 10°C min$^{-1}$, and nitrogen was used as filled gas.

#### 2.2.3.3. Differential scanning calorimetry

Differential scanning calorimetry (DSC) was used to characterize the endothermic and exothermic rates of the sample. The Q20-type differential scanning calorimeter, produced by American TA Co., Ltd, was applied to explore the heat variation of NHAWP aqueous solution during the temperature rise process. The mass of test sample was 5.1 mg, and the testing temperature ranged from 4°C to 95°C. The heat increasing rate was 5°C min$^{-1}$, and nitrogen was used as filled gas.

#### 2.2.3.4. Nuclear magnetic resonance spectroscopy

Proton nuclear magnetic resonance ($^{1}$H-NMR) spectra were recorded on a Bruker AV 400 spectrometer operating at 400 MHz. Chemical shifts are reported in parts per million (ppm) relative to the nuclear

magnetic resonance (NMR) solvent signals. The spectra were performed using $D_2O$ as solvent. By this approach, the structure of the associating polymer can be examined and the composition of each moiety which plays a crucial role in the rheological properties of polymer solution can be determined.

## 2.2.4. Viscosity of new kind of hydrophobically associating water-soluble polymer aqueous solution

To determine the thermal tackifying performance of NHAWP solution, HAAKE-type rotational viscometer was employed to test the apparent viscosity of the NHAWP aqueous solution. The concentration of NHAWP was set as 0.1%, 0.3%, 0.6% and 0.9%. The testing temperature ranges from 4°C to 90°C, which was chosen according to the real temperature of deep-water mudline and downhole environment [26]. The effect of salt concentration on the apparent viscosity of NHAWP aqueous solution was investigated by adding sodium chloride into the solution. The concentration of sodium chloride was set as 1%, 2%, 3% and 4%. Because the cement slurry is alkaline, it is necessary to evaluate the effect of pH value on the viscosity of NHAWP aqueous solution. The pH variables were adjusted to 11, 12, 13 and 14 by adding NaOH to study the alkali resistance of NHAWP.

## 2.2.5. Microscopic observation

Environmental scanning electron microscopy (ESEM) was applied to observe the state of NHAWP in aqueous solutions at both high and low temperatures. Before the ESEM test, the instant freezing method was first employed to retain the structure of NHAWP at a specific state in aqueous solutions. The preparation process of instant freezing samples was as follows: the NHAWP aqueous solution was prepared at 25°C and 60°C, respectively. And then, the NHAWP aqueous solution was taken for a drop and dropped on a pure mica sheet, which has been pre-frozen at −80°C for 3 h. After the addition of NHAWP drops, the mica sheet was frozen at −80°C for 6 h. During the freezing process, the water in the solution was removed by vacuum freeze drying. Finally, the dried samples were sprayed with gold/palladium, and the microstructure of the samples was observed and analysed by ESEM.

## 2.2.6. Evaluation of cement slurry properties

### 2.2.6.1. Preparation of cement slurry
The retarder, NHAWP and dispersant were mixed with cement. The defoamer and fluid loss additive were added to water. Cement slurry was prepared according to API 10B-2-2013. After that, all of the tests to evaluate the property of cement slurry were performed three times, and the average value was calculated and recorded.

### 2.2.6.2. Thickening time
Thickening time tests are designed to determine the length of time which slurry remains in a pumpable fluid state. The pumpability or consistency of the slurry is measured in Bearden units (Bc), a dimensionless quantity with no direct conversion factor to more common units of viscosity such as the poise. The end of a thickening time test is defined when the cement slurry reaches a consistency of 100 Bc. However, 70 Bc is generally considered to be the maximum pumpable consistency. There is a rotating cup with a fixed blade in consistometer. The cup which is driven by the motor is counterclockwise rotation at the speed of 150 r.p.m. The cement slurry in cup gives the blade a certain resistance which is proportional to the consistency of cement slurry. This resistance torque and potentiometer spring torque are in balance. Therefore, the consistency signal can be imported to the recorder through the potentiometer. Considering the temperature conditions in downhole, in our research, the consistency was tested at 75°C and 90°C and the pressure used here was 0.1 MPa.

### 2.6.2.3. American Petroleum Institute static fluid loss
According to API 10B-2-2013, a high-temperature and high-pressure fluid loss cell were employed to obtain the API fluid loss. Before testing, the cement slurry was heated and stirred continually in a rotating cup at a stirring rate of $150 \pm 15$ r.p.m. The testing temperature was set as 25°C, 50°C, 75°C and 90°C, respectively. The filtration of cement slurry was performed by passing through a 325 mesh metal sieve (with a diameter of 88.9 mm). A 6.895 MPa pressure differential was applied to the fluid

loss cell. The filtrate produced by the pressure differential was collected for 30 min. The API static fluid loss volume was twice as much as the volume of the collected filtrate.

### 2.6.2.4. Compressive strength

Cement slurries incorporating different dosages of NHAWP were prepared as standard cement stones ($5 \times 5 \times 5$ cm). After curing for 48 h, the compressive strength of hardened cement slurry was tested. The curing temperature was determined as 25°C, 50°C, 75°C and 90°C, respectively. Each cement sample was tested three times, and the average value was calculated and recorded. Tests was performed on wetted specimens with a 0.5 MPa s$^{-1}$ loading rate by WEW-300B forcing press machine, which was provided by Liaocheng Building Material Equipment Factory.

### 2.6.2.5. Rheology

A six-speed rotational viscometer was used to test the rheology of the cement slurry at 4°C, 25°C, 50°C, 75°C and 90°C, respectively. The pressure used here was 0.1 MPa. The viscometer was an electric motor-powered rotary instrument, and the rotational speed was divided into 600, 300, 200, 100, 6 and 3 r.p.m. The rotation produces shear force on the cement slurry. When the rotation speeds are the same, higher viscosity of cement slurry leads to a greater shear force. Therefore, the shear force value of the cement slurry at a certain shear rate can be indirectly characterized by measuring the rotation angle of the inner cylinder. After that, the shear rate ($\gamma$) and shear stress ($\tau$) can be calculated by equations (2.1) and (2.2).

$$\gamma = 1.705 n_r \tag{2.1}$$

and

$$\tau = 0.5099 F\theta, \tag{2.2}$$

where $n_r$ is the rotational speed of the viscometer, (unit: r.p.m.); $F$ is the spring constant; $\theta$ is the reading value of the viscometer.

# 3. Results and discussion

## 3.1. Effect of hydrophobic monomer concentration on rheological property of cement slurry

Figure 4 displays the rheological property of cement slurry with different concentrations of hydrophobic monomer, i.e. the SMA : (AM + AMPS) ratios are 0.5 : 4, 0.75 : 4, 0.85:4, 1 : 4 and 1.25 : 4. As we can see, with the decrease of hydrophobic monomer concentration, the rheological property of cement slurry first becomes stable and then becomes unstable again when the temperature changes. At different temperatures, the cement slurry with SMA : (AM + AMPS) ratio of 0.85 : 4 shows the best constant rheological performance. Therefore, SMA : (AM + AMPS) ratio of 0.85 : 4 was chosen to synthesis the NHAWP.

## 3.2. Structure analysis of new kind of hydrophobically associating water-soluble polymer

To probe the molecular structure of NHAWP, FT-IR spectra analysis were applied on the NHAWP sample, as shown in figure 5. The absorption peaks at 3342 and 3195 cm$^{-1}$ are the stretching vibration peaks of N-H in AM. The peak at 2921 cm$^{-1}$ is attributed to the antisymmetric stretching vibration of methylene. The peak at 2856 cm$^{-1}$ is due to the symmetry stretching vibration of methylene. The stretching vibration of C=O in SMA can be observed at 1731 cm$^{-1}$. The stretching vibration of C=O in AM and AMPS can be observed at 1654 cm$^{-1}$. The peak at 1612 cm$^{-1}$ is the bending vibration of N-H in AM and AMPS. The deformation vibration of methylene can be observed at 1450 cm$^{-1}$. The absorption peaks at 1188 and 1039 cm$^{-1}$ are attributed to sulfonic group. The peak at 1110 cm$^{-1}$ is attributed to C-O-C in ER-20 [27]. However, the characteristic absorption peaks of C=C cannot be found in figure 5. Therefore, it can be concluded that NHAWP was the product of the copolymerization of AM, AMPS, SMA and ER-20.

Figure 6 shows the NMR spectroscopy of NHAWP. As we can see, the chemical shift of peak a is at 3.61 and 3.67 ppm, which represents the chemical shift of H on $CH_3$-C-$CH_3$ in the AMPS structure. The H on $CH_2$ in the polymer backbone shows a chemical shift at 2.17 ppm (peak b). For peak c at 1.6 ppm, it is

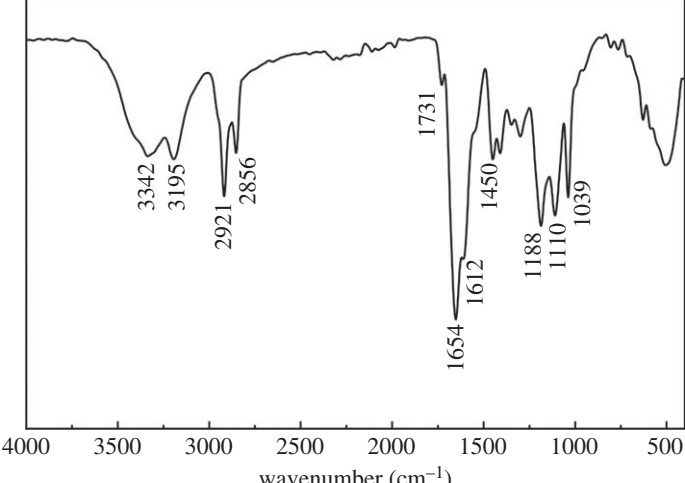

**Figure 4.** Effects of hydrophobic monomer concentration on rheological property of cement slurry.

**Figure 5.** Infrared spectrum of NHAWP.

the chemical shift of H on $CH_2$ in the long carbon chain of the non-ionic emulsifier ER-20. The peak d at 1.45 ppm is from the H on $CH_2$ in the C-O-C chain of ER-20. Where the peak at 1.15 and 1.03 ppm are from the H on $CH_2$ and $CH_3$ in hydrophobic monomer, respectively. Moreover, the peak value of C=C

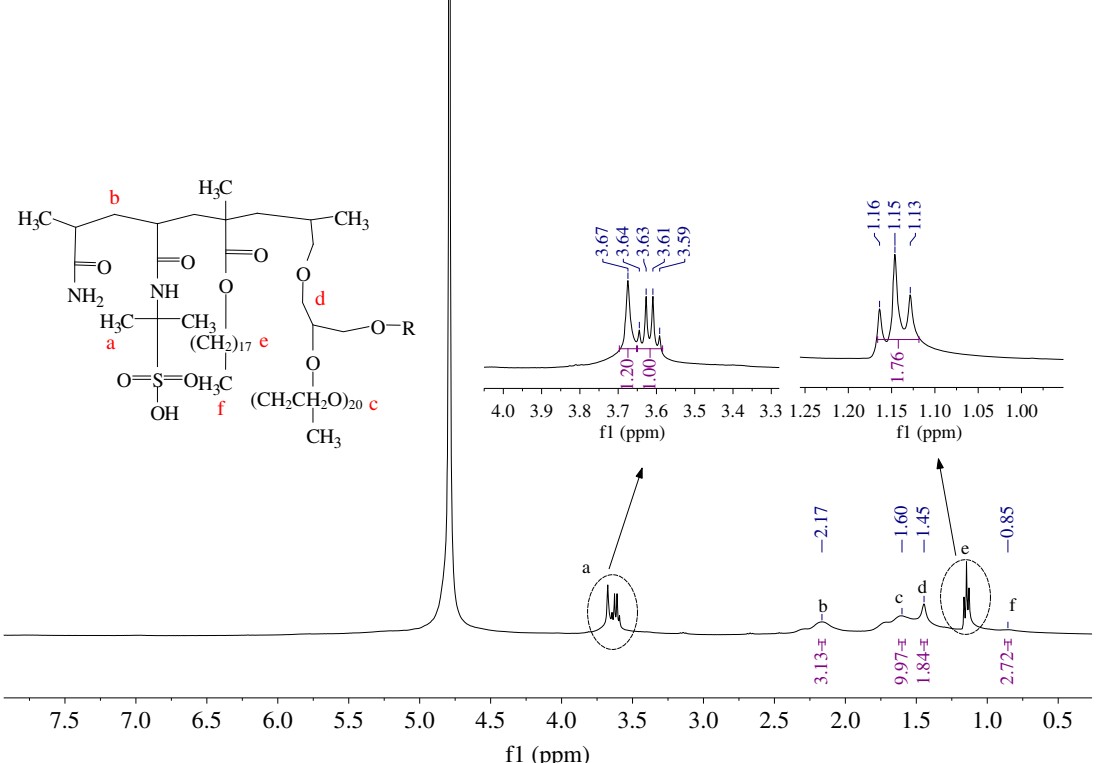

**Figure 6.** NMR spectroscopy.

cannot be identified in figure 5 (normally at 4.59 and 6.15 ppm), which illustrates that the polymer reaction is complete. This result well agrees with the FT-IR spectra analysis.

## 3.3. Thermal stability analysis of new kind of hydrophobically associating water-soluble polymer

The mass loss (TG) and mass loss rate (DTG) of NHAWP during the heating process were evaluated by thermogravimetric analysis. The results are shown in figure 7. As we can see in the TG curve in figure 7, the mass loss of NHAWP is 5.1% when the temperature increased from 30°C to 160°C. In the meantime, the rate of mass loss is low. This is mainly caused by the desorption of the adsorbed water in NHAWP. At this moment, the effective components of NHAWP were not influenced by the temperature increasing. The mass loss of NHAWP is 73.6% when the temperature increased from 310°C to 425°C. Meanwhile, the DTG curve shows that the mass loss rate is high at 371.6°C and 394.5°C. This phenomenon indicated that, when the temperature exceeds 310°C, the structure of NHAWP begins to decompose. Therefore, beyond 310°C, the NHAWP is stable, which shows an excellent high-temperature resistance property. It shall be noted that a large amount of residue can still be identified after 600°C. This is because the breaking of the molecular chain of NHAWP is a continuous process from 371.6°C, and it does not decompose completely even at 600°C.

## 3.4. Viscosity–temperature relationship of new kind of hydrophobically associating water-soluble polymer aqueous solution

### 3.4.1. Effect of new kind of hydrophobically associating water-soluble polymer concentration

The viscosity of NHAWP aqueous solution with various NHAWP concentrations (i.e. 0.1%, 0.3%, 0.6% and 0.9%) were tested at different temperatures. The results are shown in figure 8. When the temperature is less than 45°C, the apparent viscosity of NHAWP aqueous solution is basically a constant. When the temperature is higher than 45°C, the viscosity of NHAWP aqueous solution increases with the increase of temperature. This phenomenon indicates that the NHAWP does not increase the viscosity of aqueous solution at a low temperature (less than 45°C). The elevated temperature can cause the hydrophobic association of NHAWP and thereby increase the viscosity of aqueous solution. The critical association temperature of NHAWP is 45°C. For the temperature ranges

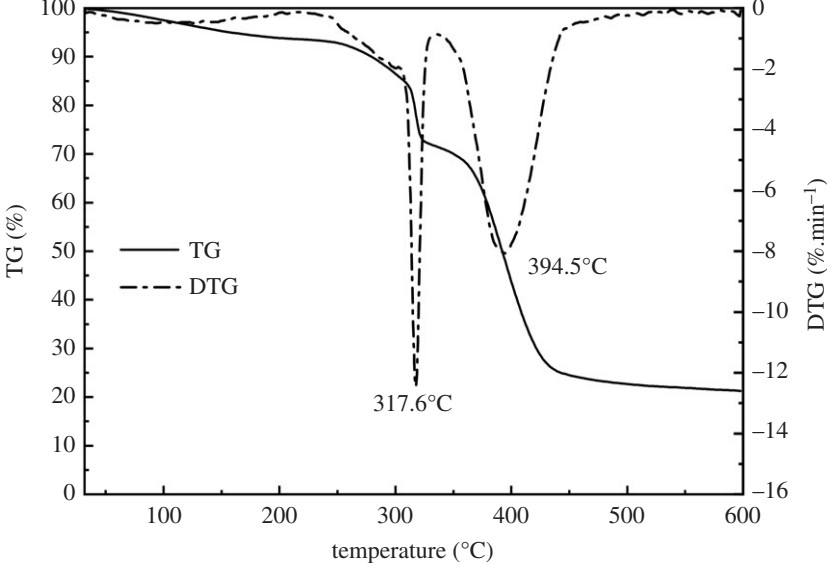

**Figure 7.** Thermogravimetric of NHAWP.

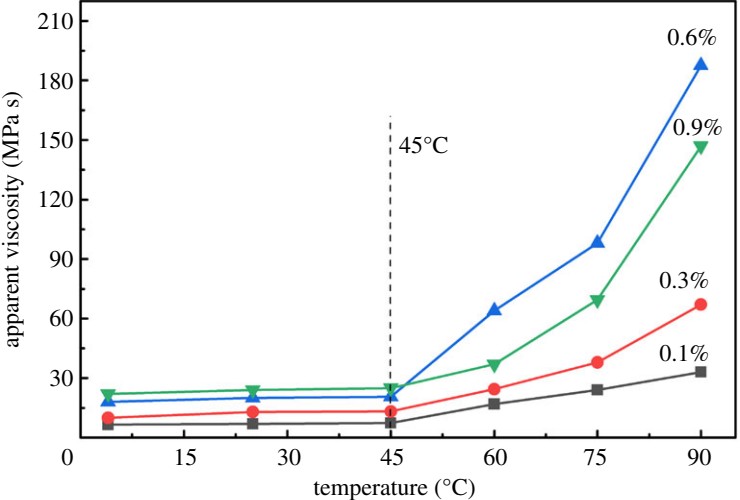

**Figure 8.** Viscosity–temperature relationship of aqueous solution with different NHAWP concentrations.

from 45°C to 90°C, with the increase of NHAWP concentration, the viscosity of NHAWP aqueous solution increases first and then decreases. When the concentration of NHAWP is 0.6%, the viscosity of aqueous solution achieves the highest value. This is due to the hydrophobic association of NHAWP consisting of intramolecular association and intermolecular association. When the concentration of NHAWP is lower than 0.3%, the intramolecular association of NHAWP is the primary reaction, which hardly affects the viscosity of the aqueous solution. When the NHAWP concentration increases to 0.6%, the intermolecular association of NHAWP occurs, which significantly increases the viscosity of the aqueous solution. However, when the concentration of NHAWP is greater than 0.6%, the solubility of NHAWP in an aqueous solution becomes limited, which shows negative effect on the hydrophobic association of NHAWP.

### 3.4.2. Effect of alkali environment

To investigate the effect of the alkali environment of cement slurry on the viscosity–temperature relationship, the viscosity test of aqueous solution with 0.6% concentration of NHAWP was performed under different pH values. The test results are shown in figure 9. We can observe that the

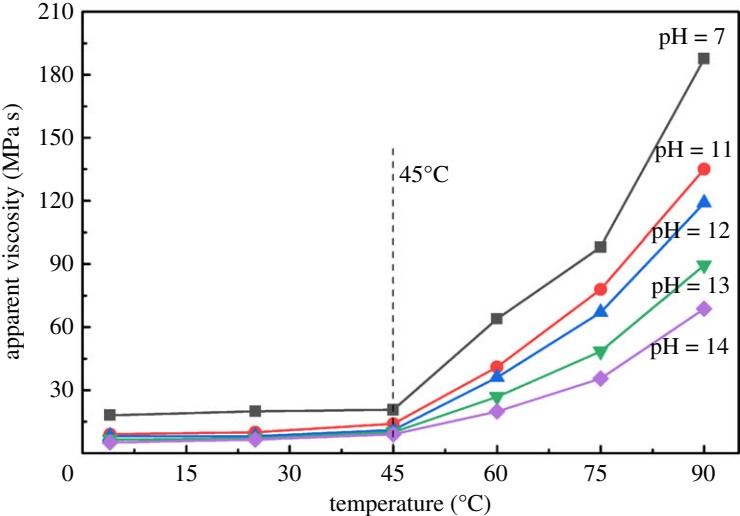

**Figure 9.** Viscosity–temperature relationship of NHAWP aqueous solution with different pH values.

critical association temperature of NHAWP is still at 45°C when the pH values vary from 7 to 14. The viscosity remains stable when temperature is lower than 45°C. Only when the temperature is higher than 45°C, the viscosity of NHAWP aqueous solution starts to increase. This phenomenon illustrated that the critical association temperature of NHAWP is independent with the variation of the alkalinity of the solution. With the increase of pH value, the viscosity of NHAWP aqueous solution decreases, which indicates that the hydrophobic association of NHAWP becomes weak when suffering from high alkalinity condition. However, when the pH value is in the range of 12 to 13, which is the pH range of cement slurry [28], NHAWP still displays obvious thermal tackifying function. Therefore, the NHAWP shows a good alkali resistance property.

### 3.4.3. Effect of salt concentration

The addition of salt led to a diminution of the viscosity of polymer solution. Muñoz-López *et al.* demonstrated the influence of salt on rheological properties of polyelectrolytes [29]. To investigate the effect of salt environment on the viscosity–temperature relationship, the viscosity test of aqueous solution with 0.6% concentration of NHAWP was performed under different salt concentration conditions. It can be seen from figure 10, for different salt concentrations, that the critical association temperature of NHAWP is still at 45°C. When the temperature is less than 45°C, the viscosity of NHAWP aqueous solution keeps unchanged. After the temperature exceeds 45°C, the viscosity of NHAWP aqueous solution with different salt concentrations begins to increase. Therefore, the critical association temperature of NHAWP is irrelevant to the changes in salt concentration. As the increase of salt concentration, the viscosity of NHAWP aqueous solution decreases. This is because the salt ions in the solution could compress the diffuse double layer to cause the molecular chain of polymer to curl, which would weaken the hydrophobic association. Even though the salt ions might have some negative effect on the hydrophobic association, NHAWP still shows excellent thermal tackifying function when the salt concentration is less than 4%.

## 3.5. Endothermic properties of new kind of hydrophobically associating water-soluble polymer aqueous solution

In order to further verify the critical association temperature of NHAWP, the DSC test was carried out on the aqueous solution with 0.6% of NHAWP. As can be seen in figure 11, the DSC curve could be divided into two sections. When the temperature range is 4°C to 45°C, the heat absorption of NHAWP solution is low. And when the temperature is higher than 45°C, the DSC curve decreases obviously, which indicates that the heat absorption of NHAWP aqueous solution increases significantly. This is because the hydrophobic association of KWHL is a massive endothermic process. The critical temperature to perform this association reaction is 45°C. Therefore, this result is consistent with the critical

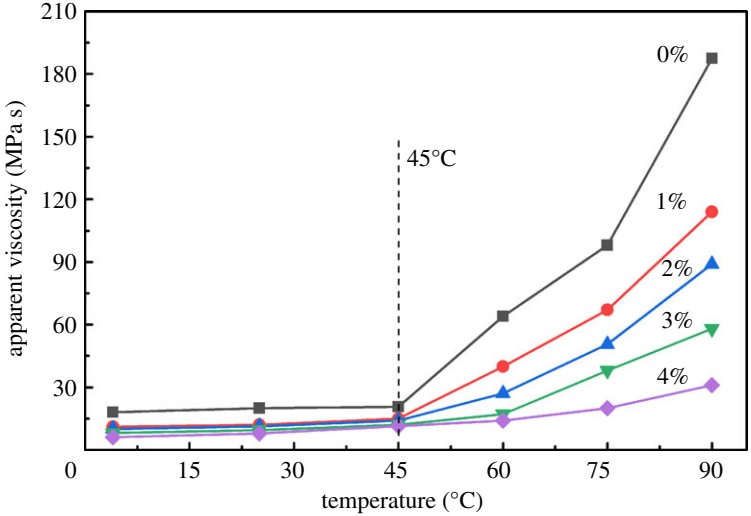

**Figure 10.** Viscosity–temperature relationship of NHAWP aqueous solution with different salt concentrations.

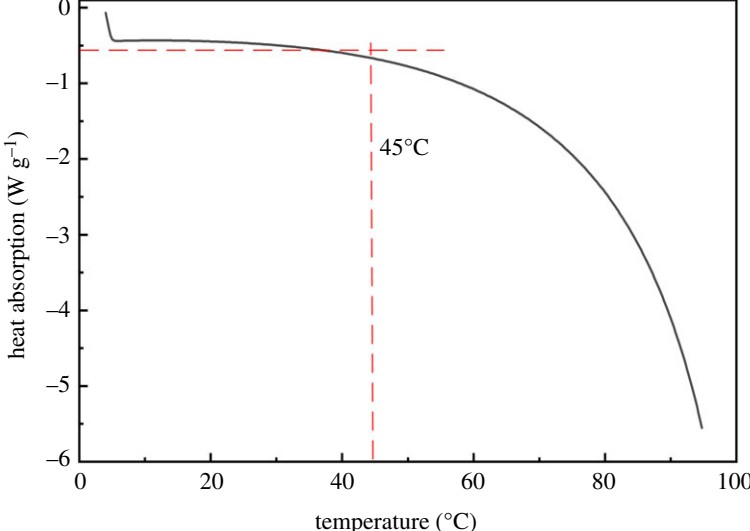

**Figure 11.** DSC spectrum of NHAWP aqueous solution.

temperature obtained via viscosity–temperature relationship tests of NHAWP aqueous solution under various conditions.

## 3.6. Microstructure of new kind of hydrophobically associating water-soluble polymer in aqueous solution

The dissolution characteristics of NHAWP at different temperatures were observed by ESEM analysis. Figure 12 is the microstructure of NHAWP in an aqueous solution tested at different temperatures. It can be seen that the molecular chain of NHAWP stretches at 60°C, which forms a spatial network structure (figure 12a,b). However, the molecular chain of NHAWP curls at 25°C and therefore forms particles with a size of tens of nanometres. When the temperature is less than 45°C, because of the curl of the molecular chain of NHAWP, the hydrophobic groups cannot associate. When the temperature is higher than 45°C, the molecular chain of NHAWP stretches accompanied by endothermic (figure 11), which is benefit for the association reaction of hydrophobic functional groups. Thus, the hydrophobic association makes the molecular chain of NHAWP-forming network structure to increase the viscosity of the NHAWP solution.

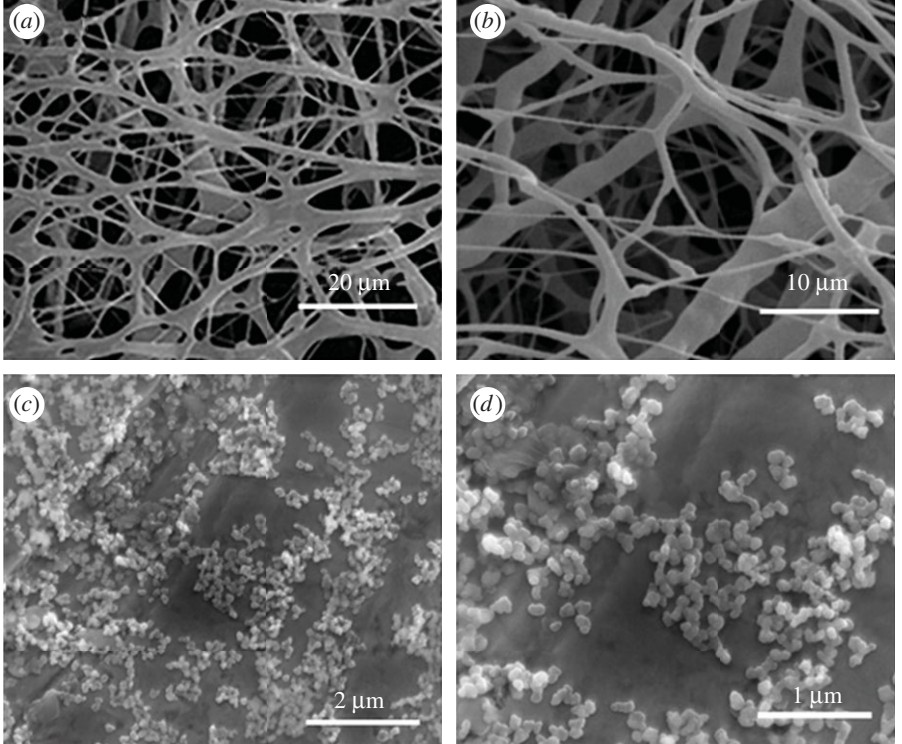

**Figure 12.** ESEM images of microstructure of NHAWP in aqueous solution ((*a*) and (*b*) are observed at 60℃; (*c*) and (*d*) are observed at 25℃).

## 3.7. Performance of cement slurry with new kind of hydrophobically associating water-soluble polymer

According to the above study, we can conclude that when the concentration of NHAWP is 0.6%, the aqueous solution shows the best viscosity–temperature relationship. Therefore, the aqueous solution with 0.6% of NHAWP was used to prepare the cement slurry samples. The type and dosage of additives in the cement slurry with NHAWP is the same as the basic cement slurry. And the water to cement ratio used here is 0.44. The effect of NHAWP on thickening property, fluid loss performance, compressive strength and rheology was evaluated in the following sections.

### 3.7.1. Effect of new kind of hydrophobically associating water-soluble polymer on thickening property

The thickening time of cement slurry is an important indicator for the safety of cementing operation [30,31]. Therefore, it is not expected that the NHAWP could influence the thickening time of the cement slurry. The thickening time of basic cement slurry and cement slurry with NHAWP was measured at different temperatures (i.e. 75℃ and 90℃), as shown in figure 13. The thickening time of basic cement slurry at 75℃ is approximately 601 min. When the temperature increases to 90℃, the thickening time of basic cement slurry reduces to 439 min. After adding NHAWP, the initial consistency of cement slurry becomes higher than the basic cement slurry, but it does not exceed 30 Bc (the initial consistency of cement slurry should be less than 30 Bc to achieve a good pumpability). The increase of initial consistency of cement slurry is mainly due to the hydrophobic association of NHAWP at high temperature (higher than 45℃). Besides, after incorporating NHAWP, the thickening time of cement slurry is prolonged, but the change is not obvious. This is due to the main molecular chain of NHAWP that is polymerized by AM and AMPS. The AM can hydrolyse carboxyl groups in strong alkaline environment of the cement slurry. Meanwhile, AMPS has sulfonic acid groups, which can adsorb calcium ions in cement slurry. However, after the polymerization of AM and AMPS, only little amount of AM and AMPS remain in the cement slurry. Therefore, as expected, the NHAWP shows little effect on the thickening property of the cement slurry.

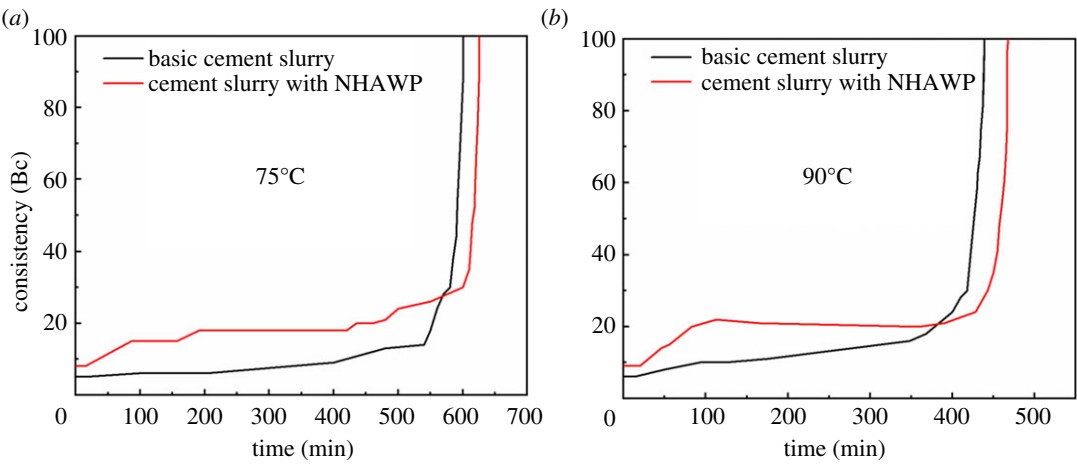

**Figure 13.** Thickening curve of basic cement slurry and cement slurry with NHAWP.

**Table 2.** Fluid loss of cement slurry.

| temperature (°C) | basic cement slurry (ml) | cement slurry with NHAWP (ml) |
| --- | --- | --- |
| 25 | 48 | 43 |
| 50 | 46 | 36 |
| 75 | 44 | 31 |
| 90 | 48 | 38 |

### 3.7.2. Effect of new kind of hydrophobically associating water-soluble polymer on fluid loss performance

The fluid loss performance of basic cement slurry and cement slurry with NHAWP was evaluated at different temperatures. The results are shown in table 2. It can be seen from table 2 that the fluid loss of cement slurry with NHAWP is less than that of basic cement slurry, which illustrates that the NHAWP displays positive effect on reducing the fluid loss of cement slurry, especially at temperatures higher than 45°C. This is attributed to the hydrophobic association of NHAWP increasing the viscosity of cement slurry and thereby increasing the flow resistance of free water in cement slurry.

### 3.7.3. Effect of new kind of hydrophobically associating water-soluble polymer on compressive strength

The influence of NHAWP on the compressive strength of hardened cement was investigated at different temperatures (i.e. 4°C, 25°C, 50°C, 75°C and 90°C). As can be seen in figure 14, the NHAWP shows almost no effect on the compressive strength of hardened cement at 4°C and 25°C. However, when the temperature increases to more than 50°C, the NHAWP shows a certain negative effect on the compressive strength. This is because the molecular chain of NHAWP begins to stretch when the temperature exceeds 45°C. The hydrophilic groups in the molecular chain of NHAWP attaches to the surface of cement particles, thus affecting the development of compressive strength of cement, which acts as a retarder. Although the NHAWP reduces the compressive strength of hardened cement, the strength of hardened cement still can meet the requirements of oil well cementing (the strength of hardened cement needs to reach 13.8 MPa [32]).

### 3.7.4. Effect of new kind of hydrophobically associating water-soluble polymer on rheology

The rheological properties of the basic cement slurry and cement slurry with NHAWP were studied at different temperatures. The results are shown in figure 15. The original viscometer parameters for calculating the rheological curve are provided in tables 3 and 4. The viscosity of basic cement slurry decreases with the increase in temperature. Meanwhile, the rheological property of cement slurry varies greatly at different temperatures. However, the rheological curves of cement slurry with NHAWP almost show the same variation at different temperatures. And the rise of temperature

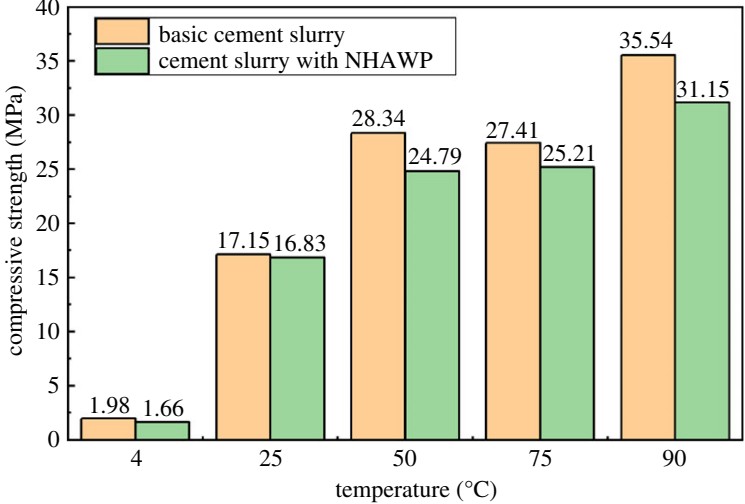

**Figure 14.** The compressive strength of hardened cement.

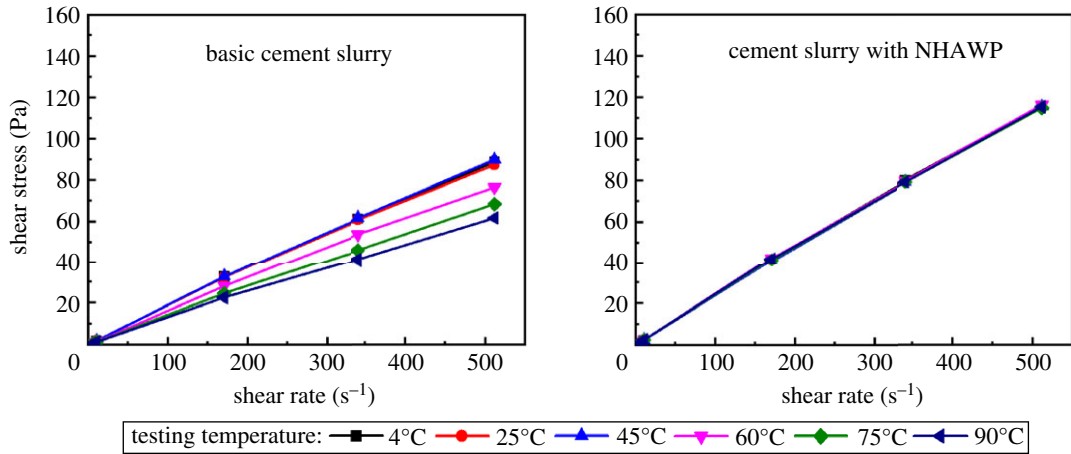

**Figure 15.** Rheological curve of cement slurry with and without NHAWP at different temperatures.

**Table 3.** Viscometer parameters for basic cement slurry.

| temperature (°C) | $\Phi_{300}$ | $\Phi_{200}$ | $\Phi_{100}$ |
|---|---|---|---|
| 4 | 173 | 120 | 65 |
| 25 | 173 | 118 | 63 |
| 45 | 175 | 120 | 62 |
| 60 | 149 | 106 | 55 |
| 75 | 133 | 88 | 47 |
| 90 | 120 | 82 | 43 |

**Table 4.** Viscometer parameters for cement slurry with NHAWP.

| temperature (°C) | $\Phi_{300}$ | $\Phi_{200}$ | $\Phi_{100}$ |
|---|---|---|---|
| 4 | 225 | 155 | 80 |
| 25 | 224 | 152 | 78 |
| 45 | 225 | 154 | 78 |
| 60 | 222 | 151 | 77 |
| 75 | 223 | 153 | 79 |
| 90 | 226 | 155 | 81 |

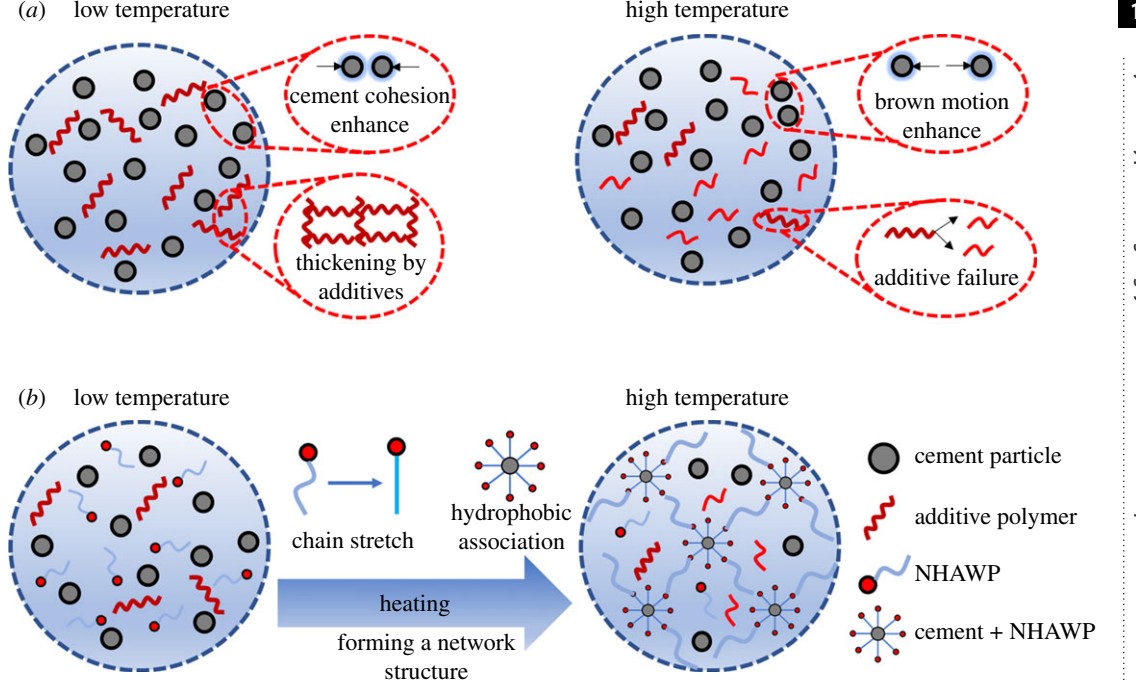

**Figure 16.** Reaction mechanism of NHAWP in cement slurry.

nearly displays no influence on the viscosity of cement slurry. This phenomenon indicates that the cement slurry with NHAWP shows a good constant rheological property at different temperatures.

## 3.8. Discussion of reaction mechanism of new kind of hydrophobically associating water-soluble polymer in cement slurry

There are two main mechanisms to enhance the apparent viscosity of cement slurry at low temperatures: first, low temperature leads to the improvement of the cohesion force of cement particles [33]. The other reason is that the polymers in cement additives (e.g. retarder and fluid loss agent) increase the viscosity of the cement slurry at low temperatures [34], as shown in figure 16a, low temperature. When the temperature increases, the Brownian motion of cement particles enhances, which leads to the reduction of cohesion force [35]. Meanwhile, high temperature results in the breakdown or increased dispersion of the additives, thereby reducing the viscosity of the cement slurry (figure 16a, high temperature). After the incorporation of NHAWP, the NHAWP shows good dispersion at low temperatures, therefore the viscosity of cement slurry is similar to that of basic cement slurry (see figures 15 and 16b, low temperature). When the temperature rises to more than 45°C, the chain of NHAWP stretches due to heating. After that, the NHAWP performs hydrophobic association with cement particles to form a network structure in cement matrix (figures 12 and 16b, high temperature). Therefore, the shear stress–shear rate relationship of NHAWP-enhanced cement slurry is nearly unchanged with the variation of temperature (figure 12).

## 4. Conclusion

A NHAWP, with the function of thermal tackifying, was designed and developed to make a constant rheology cement slurry in the temperature range of 4–90°C. According to the experimental test and theoretical analysis, the following conclusions can be drawn:

(1) NHAWP was developed by copolymerization of hydrophilic monomers and hydrophobic monomers. According to the analysis of viscosity–temperature relationship of NHAWP aqueous solution, the critical association temperature of NHAWP is determined, which is 45°C. This critical association temperature is independent of NHAWP concentration, salt concentration and alkalinity of solution.

(2) When the temperature is less than 45°C, the molecule chain of NHAWP curls and shows little influence on the viscosity of the solution. However, when the temperature is higher than 45°C, the molecule chain of NHAWP stretches to induce the hydrophobic association. The intermolecular hydrophobic association of NHAWP could form spatial network structure, which increases the viscosity of NHAWP aqueous solution.

(3) The developed NHAWP shows good thermal stability, as well as excellent salt and alkali resistance properties. Furthermore, the NHAWP displays almost no negative influence on the thickening time and compressive strength of cement slurry. And the fluid loss of cement slurry was reduced after incorporating NHAWP. Therefore, the NHAWP can be used as a constant rheology agent to prepare a cement slurry with constant rheology in the temperature range of 4–90°C.

Data accessibility. The data is available at the Dryad Digital Repository (https://doi.org/10.5061/dryad.mcvdnck1n).

Authors' contributions. Y.B.: conceptualization, data curation, funding acquisition, investigation, methodology, resources, supervision and writing—original draft; M.X.: formal analysis, investigation, methodology and writing—original draft; H.L.: conceptualization, investigation, methodology, supervision and writing—original draft; A.Z.: investigation, supervision and writing—review and editing; J.D.: data curation, investigation, methodology, writing—original draft and writing—review and editing; X.Y.: data curation, investigation and methodology; S.G.: conceptualization, methodology and supervision.

All authors gave final approval for publication and agreed to be held accountable for the work performed therein.

Competing interests. The authors declare that they have no conflict of interest.

Funding. We would like to thank to Natural Science Foundation of China (grant no. 51974355), the Fundamental Research Funds for the Central Universities (grant no. 18CX02161A), Major scientific and technological projects of CNPC (grant no. ZD2019-184-003), Shandong Provincial Postdoctoral Science Foundation (grant no. 201702025) and Program for Changjiang Scholars and Innovative Research Team in University (grant no. IRT1086) for their support.

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
