## [Peer Review File · Royal Society Open Science]

Review History

RSOS-211170.R0 (Original submission)

Review form: Reviewer 1

Is the manuscript scientifically sound in its present form?

Yes

Are the interpretations and conclusions justified by the results?

Yes

Is the language acceptable?

Yes

Do you have any ethical concerns with this paper?

No

Have you any concerns about statistical analyses in this paper?

No

Recommendation?

Major revision is needed (please make suggestions in comments)

Comments to the Author(s)

The manuscript entitled “A novel hydrophobically associating water-soluble polymer utilized as constant rheology agent for cement slurry” reports the synthesis of a novel HAWP and its use as an additive in the oil well cement slurry. In general, the manuscript is well written and described an important application area. Authors reported only one polymer without mention the influence of hydrophobic moieties in the physico-chemical properties of the resulting material.

Furthermore, I detect the absence of specific characterization methods for confirming the structure of final polymer. For instance, my recommendation is major revision and I have listed my comments which need addressing below.

Major revision

1. Authors reported only one reaction from “inverse microemulsion polymerization”. Why? Can the authors include the molar ratio of reactants in the experimental section or as a new Table? They expected the emulsifier (Er-20) acts as a hydrophobic monomer. What is the Kp of Er-20? Can it polymerize at same time than SMA?
2. FT-IR analysis only reveals the chemical bonds of compounds. Why they did not perform NMR analyses using DMSO as solvent or a mixture (DMSO/Water). This technique not only would allow to examine the structure of the associating polymer, but also determine the composition of each moiety which play a crucial role in the rheological properties of polymer solution.
3. The molecular weight is an important parameter in the thickening behavior of HAWP. Can the authors determine or expect the Mn of KWHL-1? Why they use only one concentration of hydrophobic monomer(s)? Then, they cannot explain the influence of hydrophobic moieties if they did not performed additional experiments.
4. Authors reported the Dynamic Scanning Calorimetry (DSC) measurement in the experimental section. Why they did not reported the Tg of resulting polymer? Based on the inverse microemulsion polymerization process, they can access to a polymer with both moieties: hydrophilic backbone and rich hydrophobic moiety.
5. The main purpose of this work is to evaluate the impact of HAWP in the cement slurry. However, results presented in the section 3.6.1 are weak. The authors related that the addition of KWHL-2 demonstrates little effects on the thickening property of the cement slurry. Can they explain this behavior?

Minor revision

- 1) Abstract section. The abbreviation KWHL-1 is neither appropriate nor definite.
- 2) Can they definite KWHL-1 which corresponds to synthesize polymer reported in this work?
- 3) Both Françoise Candau and Enrique Jiménez-Regalado are reported a lot of investigations related to the HAWP. Can they consider in this work?
- 4) Can consider the emulsifier Er-20 as a monomer? See the section 2.2.3.1
- 5) Themogravimetric analysis was performed at a temperature range from 30 to 600 oC. The Fig 5 revealed a large amount of residue (ca. to 20 wt.%) after 600 oC. Can it attribute to a specific compound of KWHL-1?

Results and discussion

- 6) Authors reported that the absorption peak of C=C could not be found. What kind of peak expected after polymerization process?
- 7) The addition of salt led to a diminution of the viscosity of polymer solution. Recently, Cesar Muñoz-López demonstrated the influence of salt on rheological properties of polyelectrolytes (J of Applied Polym. Sci., 2021, 138, 43, 51270).
- 8) The process reported in this work is it reproducible?

Review form: Reviewer 2

Is the manuscript scientifically sound in its present form?

Yes

Are the interpretations and conclusions justified by the results?

No

Is the language acceptable?

Yes

Do you have any ethical concerns with this paper?

No

Have you any concerns about statistical analyses in this paper?

Yes

Recommendation?

Major revision is needed (please make suggestions in comments)

Comments to the Author(s)

This topic is very interesting. As the author said, a constant rheology agent is very important for the cementing engineering of oil and natural gas wells.

But this paper contain some unconvincing data, authors need provide more experimental procedure and original experimental data.

In this work, authors used the monomer (SMA, AMPS, AM) to develop a new hydrophobically associating water-soluble polymer (KWHL-1), and they studied the effects of the synthetic polymer on the consistency, compressive strength, and rheological properties of cement slurry. From the results of the manuscript, the authors are proving that KWHL-1 is beneficial to improve the performance of cement slurry. However, combining the results of consistency and rheological properties of the cement slurries, which confused me. According to the consistency results, the consistency of cement slurry with KWHL-1 increased with temperature rise, but the rheological properties of the cement slurry have no change with increasing temperature. The two results are inconsistent.

Additionally, the rheological properties of the cement slurry are too excellent to believe. Authors need provide more experimental procedure and original experimental data.

Decision letter (RSOS-211170.R0)

Dear Dr Liu:

Title: A novel hydrophobically associating water-soluble polymer utilized as constant rheology agent for cement slurry
Manuscript ID: RSOS-211170

The editor assigned to your manuscript has now received comments from reviewers. We would like you to revise your paper in accordance with the referee and Subject Editor suggestions which can be found below (not including confidential reports to the Editor). Please note this decision does not guarantee eventual acceptance.

Please submit your revised paper before 10-Dec-2021. Please note that the revision deadline will expire at 00.00am on this date. If we do not hear from you within this time then it will be assumed that the paper has been withdrawn. In exceptional circumstances, extensions may be possible if agreed with the Editorial Office in advance. We do not allow multiple rounds of revision so we urge you to make every effort to fully address all of the comments at this stage. If deemed necessary by the Editors, your manuscript will be sent back to one or more of the original reviewers for assessment. If the original reviewers are not available we may invite new reviewers.

Please also include the following statements alongside the other end statements. As we cannot publish your manuscript without these end statements included, if you feel that a given heading is not relevant to your paper, please nevertheless include the heading and explicitly state that it is not relevant to your work.

- Ethics statement

Please clarify whether you received ethical approval from a local ethics committee to carry out your study. If so please include details of this, including the name of the committee that gave consent in a Research Ethics section after your main text. Please also clarify whether you received informed consent for the participants to participate in the study and state this in your Research Ethics section.

OR

Please clarify whether you obtained the necessary licences and approvals from your institutional animal ethics committee before conducting your research. Please provide details of these licences and approvals in an Animal Ethics section after your main text.

OR

Please clarify whether you obtained the appropriate permissions and licences to conduct the fieldwork detailed in your study. Please provide details of these in your methods section.

- Data accessibility

It is a condition of publication that you make available the data and research materials supporting the results in the article. Datasets should be deposited in an appropriate publicly available repository and details of the associated accession number, link or DOI to the datasets must be included in the Data Accessibility section of the article

(<https://royalsocietypublishing.org/rsos/for-authors#question17>). Reference(s) to datasets should also be included in the reference list of the article with DOIs (where available).

Please include a Data Availability section after your main text stating where supporting data are available from, or where they will be made available should your article be accepted for publication.

<http://datadryad.org/submit?journalID=RSOS&manu=RSOS-211170>

- Competing interests

Please include a Competing Interests section after your main text declaring any financial or non-financial competing interests. If you have no competing interests please state 'I/we have no competing interests.'

- Authors' contributions

Please include an Authors' Contributions section at the end of your main text detailing the contribution of each author. All authors should have read and approved the manuscript before submission and this should be stated in the Authors' Contributions section.

The list of Authors should meet all of the following criteria; 1) substantial contributions to conception and design, or acquisition of data, or analysis and interpretation of data; 2) drafting the article or revising it critically for important intellectual content; and 3) final approval of the version to be published.

- Acknowledgements

- Funding statement

Please include a funding section after your main text which lists the source of funding for each author.

Yours sincerely,
Dr Ellis Wilde
Publishing Editor, Journals

On behalf of the Subject Editor Professor Anthony Stace and the Associate Editor Professor Chaohua Cui.

RSC Associate Editor
Comments to the Author:
(There are no comments.)

RSC Subject Editor
Comments to the Author:
(There are no comments.)

Reviewers' Comments to Author:

Reviewer: 1

Comments to the Author(s)

The manuscript entitled "A novel hydrophobically associating water-soluble polymer utilized as constant rheology agent for cement slurry" reports the synthesis of a novel HAWP and its use as an additive in the oil well cement slurry. In general, the manuscript is well written and described an important application area. Authors reported only one polymer without mention the influence of hydrophobic moieties in the physico-chemical properties of the resulting material. Furthermore, I detect the absence of specific characterization methods for confirming the structure of final polymer. For instance, my recommendation is major revision and I have listed my comments which need addressing below.

Major revision

1. Authors reported only one reaction from "inverse microemulsion polymerization". Why? Can the authors include the molar ratio of reactants in the experimental section or as a new Table? They expected the emulsifier (Er-20) acts as a hydrophobic monomer. What is the K_p of Er-20? Can it polymerize at same time than SMA?

2. FT-IR analysis only reveals the chemical bonds of compounds. Why they did not perform NMR analyses using DMSO as solvent or a mixture (DMSO/Water). This technique not only would allow to examine the structure of the associating polymer, but also determine the composition of each moiety which play a crucial role in the rheological properties of polymer solution.

3. The molecular weight is an important parameter in the thickening behavior of HAWP. Can the authors determine or expect the M_n of KWHL-1? Why they use only one concentration of hydrophobic monomer(s)? Then, they cannot explain the influence of hydrophobic moieties if they did not performed additional experiments.

4. Authors reported the Dynamic Scanning Calorimetry (DSC) measurement in the experimental section. Why they did not reported the T_g of resulting polymer? Based on the inverse microemulsion polymerization process, they can access to a polymer with both moieties: hydrophilic backbone and rich hydrophobic moiety.

5. The main purpose of this work is to evaluate the impact of HAWP in the cement slurry. However, results presented in the section 3.6.1 are weak. The authors related that the addition of KWHL-2 demonstrates little effects on the thickening property of the cement slurry. Can they explain this behavior?

Minor revision

1) Abstract section. The abbreviation KWHL-1 is neither appropriate nor definite.

2) Can they definite KWHL-1 which corresponds to synthesize polymer reported in this work?

3) Both Françoise Candau and Enrique Jiménez-Regalado are reported a lot of investigations related to the HAWP. Can they consider in this work?

4) Can consider the emulsifier Er-20 as a monomer? See the section 2.2.3.1

5) Themogravimetric analysis was performed at a temperature range from 30 to 600 oC. The Fig 5 revealed a large amount of residue (ca. to 20 wt.%) after 600 oC. Can it attribute to a specific compound of KWHL-1?

Results and discussion

6) Authors reported that the absorption peak of C=C could not be found. What kind of peak expected after polymerization process?

7) The addition of salt led to a diminution of the viscosity of polymer solution. Recently, Cesar Muñoz-López demonstrated the influence of salt on rheological properties of polyelectrolytes (J of Applied Polym. Sci., 2021, 138, 43, 51270).

8) The process reported in this work is it reproducible?

Reviewer: 2

Comments to the Author(s)

This topic is very interesting. As the author said, a constant rheology agent is very important for the cementing engineering of oil and natural gas wells.

But this paper contain some unconvincing data, authors need provide more experimental procedure and original experimental data.

In this work, authors used the monomer (SMA, AMPS, AM) to develop a new hydrophobically associating water-soluble polymer (KWHL-1), and they studied the effects of the synthetic polymer on the consistency, compressive strength, and rheological properties of cement slurry. From the results of the manuscript, the authors are proving that KWHL-1 is beneficial to improve the performance of cement slurry. However, combining the results of consistency and rheological properties of the cement slurries, which confused me. According to the consistency results, the consistency of cement slurry with KWHL-1 increased with temperature rise, but the rheological properties of the cement slurry have no change with increasing temperature. The two results are inconsistent.

Additionally, the rheological properties of the cement slurry are too excellent to believe. Authors need provide more experimental procedure and original experimental data.

Author's Response to Decision Letter for (RSOS-211170.R0)

See Appendix A.

RSOS-211170.R1 (Revision)

Review form: Reviewer 2

Is the manuscript scientifically sound in its present form?

Yes

Are the interpretations and conclusions justified by the results?

No

Is the language acceptable?

Yes

Do you have any ethical concerns with this paper?

No

Have you any concerns about statistical analyses in this paper?

No

Recommendation?

Accept as is

Comments to the Author(s)

This manuscript can be accepted.

Decision letter (RSOS-211170.R1)

Dear Dr Liu:

Title: A novel hydrophobically associating water-soluble polymer utilized as constant rheology agent for cement slurry
Manuscript ID: RSOS-211170.R1

It is a pleasure to accept your manuscript in its current form for publication in Royal Society Open Science. The chemistry content of Royal Society Open Science is published in collaboration with the Royal Society of Chemistry.

Yours sincerely,
Dr Ellis Wilde
Publishing Editor, Journals

On behalf of the Subject Editor Professor Anthony Stace and the Associate Editor Professor Chaohua Cui.

RSC Associate Editor: 1
Comments to the Author:
(There are no comments.)

RSC Associate Editor: 2
Comments to the Author:
(There are no comments.)

Reviewer(s)' Comments to Author:
Reviewer: 3

Comments to the Author(s)
This manuscript can be accepted.

Appendix A

**Authors' Reply to the reviewers' comments on
A novel hydrophobically associating water-soluble polymer utilized as constant
rheology agent for cement slurry
(Manuscript ID: RSOS-211170)**

We would like to thank the editor/reviewers for the effort and time spent on reviewing our paper. The comments are generally insightful and constructive for improving the paper. We have tried to incorporate them in the revised manuscript as much as we think appropriate.

Reviewers' Comments to Author:

Reviewer: 1

Recommendation: *The manuscript entitled "A novel hydrophobically associating water-soluble polymer utilized as constant rheology agent for cement slurry" reports the synthesis of a novel HAWP and its use as an additive in the oil well cement slurry. In general, the manuscript is well written and described an important application area. Authors reported only one polymer without mention the influence of hydrophobic moieties in the physico-chemical properties of the resulting material. Furthermore, I detect the absence of specific characterization methods for confirming the structure of final polymer. For instance, my recommendation is major revision and I have listed my comments which need addressing below.*

Authors' reply: Thank you very much for your time and effort to comment our manuscript. All the page and line number mentioned in Author's reply are regarding the "manuscript with changes marked".

Q1. *Authors reported only one reaction from "inverse microemulsion polymerization". Why? Can the authors include the molar ratio of reactants in the experimental section or as a new Table? They expected the emulsifier (Er-20) acts as a hydrophobic monomer. What is the Kp of Er-20? Can it polymerize at same time than SMA?*

Authors' reply: Many thanks for your comments. The emulsifier (Er-20) can also act as a hydrophobic monomer, which can polymerize at the same time as SMA due to it possesses the C=C bond. This has been reported in section 2.2.1.

Please see page 6, line 155-156.

Because the Er-20 is a kind of non-ionic emulsifier, it does not possess Kraft point (Kp).

The mole ratio of AM and AMPS is 9:1, which are provided in section 2.2.2.

Please see page 6-7, line 163-171.

Revisions have been made in the manuscript:

(See page 6, line 155-156)

The final molecular structure of developed NHAWP is shown in Fig. 2d. Moreover, the emulsifier (Er-20) can also act as a hydrophobic monomer, which can polymerize at the same time as SMA due to it possesses the C=C bond.

(See page 6-7, line 163-171)

The synthesis procedure is shown in Fig. 3 and the detailed procedures are as follows: firstly, 100 g distilled water was added into a beaker. After that, AM and AMPS with the mass ratio of 3:1 (i.e., mole ratio of 9:1) were added into the beaker.

Q2. FT-IR analysis only reveals the chemical bonds of compounds. Why they did not perform NMR analyses using DMSO as solvent or a mixture (DMSO/Water). This technique not only would allow to examine the structure of the associating polymer, but also determine the composition of each moiety which play a crucial role in the rheological properties of polymer solution.

Authors' reply: Many thanks for your comments. The NMR analyses have been performed to better characterize the structure of the polymer. Fig. 6 shows the NMR spectroscopy of NHAWP. As we can see, the chemical shift of peak a is at 3.61 and 3.67 ppm, which represents the chemical shift of H on CH₃-C-CH₃ in the AMPS structure. The H on CH₂ in the polymer backbone shows a chemical shift at 2.17 ppm (peak b). For peak c at 1.6 ppm, it is the chemical shift of H on CH₂ in the long carbon chain of the non-ionic emulsifier ER-20. The peak d at 1.45 ppm is from the H on CH₂ in the C-O-C chain of ER-20. Where the peak at 1.15 and 1.03 ppm are from the H on CH₂ and CH₃ in hydrophobic monomer respectively. Moreover, the peak value of C=C cannot be identified in Fig. 5 (normally at 4.59 and 6.15 ppm), which illustrates that the polymer reaction is complete. This result well agrees with the FTIR spectra analysis.

Fig. 6 NMR spectroscopy

Please see page 8, line 210-215 and page 13, line 332-338.

Revisions have been made in the manuscript:

(see page 8, line 210-215)

2.2.3.4 Nuclear magnetic resonance (NMR) spectroscopy

Proton nuclear magnetic resonance ($^1\text{H-NMR}$) spectra were recorded on a Bruker AV 400 spectrometer operating at 400 MHz. Chemical shifts are reported in parts per million (ppm) relative to the NMR solvent signals. The spectra were performed using D_2O as solvent. By this approach, the structure of the associating polymer can be examined and the composition of each moiety which play a crucial role in the rheological properties of polymer solution can be determined.

(see page 13, line 332-338)

Fig. 6 shows the NMR spectroscopy of NHAWP. As we can see, the chemical shift of peak a is at 3.61 and 3.67 ppm, which represents the chemical shift of H on $\text{CH}_3\text{-C-CH}_3$ in the AMPS structure. The H on CH_2 in the polymer backbone shows a chemical shift at 2.17 ppm (peak b). For peak c at 1.6 ppm, it is the chemical shift of H on CH_2 in the long carbon chain of the no-ionic emulsifier ER-20. The peak d at 1.45 ppm is from the H on CH_2 in the C-O-C chain of ER-20. Where the peak at 1.15 and 1.03 ppm are from the H on CH_2 and CH_3 in hydrophobic monomer respectively. Moreover, the peak value of C=C cannot be identified in Fig. 5 (normally at 4.59 and 6.15 ppm), which illustrates that the polymer reaction is complete. This result well agrees with the FTIR spectra analysis.

Q3. *The molecular weight is an important parameter in the thickening behavior of HAWP. Can the authors determine or expect the M_n of KWHL-1? Why they use only one concentration of hydrophobic monomer(s)? Then, they cannot explain the influence of hydrophobic moieties if they did not perform additional experiments.*

Authors' reply: Many thanks for your comments. The M_n and M_w of NHAWP are 6259 and 6727 respectively. This result shows that the NHAWP is a narrow molecular weight distribution polymer ($M_w:M_n = 1.065$) with stable performance. This has been clarified in section 2.2.2. Please see page 7, line 182-184.

The mass ratio of hydrophobic monomer and hydrophilic monomer has been pre-adjusted. The effect of hydrophobic monomer on rheological property of cement was also studied in section 3.1. Fig. 4 displays the rheological property of cement slurry with different concentration of hydrophobic monomer, i.e., the SMA: (AM+AMPS) ratios are 0.5:4, 0.75:4, 0.85:4, 1:4 and 1.25:4. As we can see, with the decrease of hydrophobic monomer concentration, the rheological property of cement slurry first becomes stable and then becomes unstable again when the temperature changes. At different temperatures, the cement slurry with SMA: (AM+AMPS) ratio

of 0.85:4 shows the best constant rheological performance. Therefore, SMA: (AM+AMPS) ratio of 0.85:4 was chosen to synthesis the NHAWP.

Fig. 4 Effects of hydrophobic monomer concentration on rheological property of cement slurry.

Please see page 7, line 176-178 and page 11-12, line 303-311.

Revisions have been made in the manuscript:

(see page 7, line 182-184)

The Mn and Mw of NHAWP are 6259 and 6727 respectively. This result shows that the NHAWP is a narrow molecular weight distribution polymer (Mw:Mn = 1.065) with stable performance.

(see page 7, line 176-178)

The dosage of non-ionic reactive emulsifier and SMA were 21.5% and 25% by the total mass of AM and AMPS respectively. It shall be noted that the mass ratio of hydrophobic monomer and hydrophilic monomer has been pre-adjusted. The effect of hydrophobic monomer on rheological property of cement was also studied in section 3.1.

(see page 11-12, line 303-311)

3.1 Effect of hydrophobic monomer concentration on rheological property of cement slurry

Fig. 4 displays the rheological property of cement slurry with different concentration of hydrophobic monomer, i.e., the SMA: (AM+AMPS) ratios are 0.5:4, 0.75:4, 0.85:4, 1:4 and 1.25:4. As we can see, with the decrease of hydrophobic monomer concentration, the rheological property of cement slurry first becomes stable and then becomes unstable again when the temperature changes. At different temperatures, the cement slurry with SMA: (AM+AMPS) ratio of 0.85:4 shows the best constant rheological performance. Therefore, SMA: (AM+AMPS) ratio of 0.85:4 was chosen to synthesis the NHAWP.

Q4. Authors reported the Dynamic Scanning Calorimetry (DSC) measurement in the experimental section. Why did they not report the TG of resulting polymer? Based on the inverse microemulsion polymerization process, they can access to a polymer with both moieties: hydrophilic backbone and rich hydrophobic moiety.

Authors' reply: Many thanks for your comments. The TG results have been reported in section 3.3. As we can see in the TG curve in Fig. 7, the mass loss of NHAWP is 5.1% when the temperature increased from 30 to 160°C. This mainly caused by the desorption of the adsorbed water in NHAWP. The mass loss of NHAWP is 73.6% when the temperature increased from 310 to 425°C. This phenomenon indicated that, when the temperature exceeds 310°C, the structure of NHAWP begins to decompose.

Fig. 7 Thermogravimetric of NHAWP.

Please see page 14-15, line 346-366.

Revisions have been made in the manuscript:

(see page 14-15, line 346-366)

The mass loss (TG) and mass loss rate (DTG) of NHAWP during the heating process were evaluated by thermogravimetric analysis. The results are shown in Fig. 7. As we can see in the TG curve in Fig. 7, the mass loss of NHAWP is 5.1% when the temperature increased from 30 to 160°C. In the meantime, the rate of mass loss is low. This mainly caused by the desorption of the adsorbed water in NHAWP. At this moment, the effective components of NHAWP were not influenced by the temperature increasing. The mass loss of NHAWP is 73.6% when the

temperature increased from 310 to 425°C. Meanwhile, the DTG curve shows that the mass loss rate is high at 371.6°C and 394.5°C. This phenomenon indicated that, when the temperature exceeds 310°C, the structure of NHAWP begins to decompose. Therefore, beyond 310°C, the NHAWP is stable, which shows an excellent high temperature resistance property. It shall be noted that a large amount of residue can still be identified after 600°C. This is because the broken of molecular chain of NHAWP is a continuous process from 371.6°C, and it does not decompose completely even at 600°C.

Q5. The main purpose of this work is to evaluate the impact of HAWP in the cement slurry. However, results presented in the section 3.6.1 are weak. The authors related that the addition of KWHL-2 demonstrates little effects on the thickening property of the cement slurry. Can they explain this behavior?

Authors' reply: Many thanks for your comments. After incorporating NHAWP, the thickening time of cement slurry is prolonged, but the change is not obvious. This is due to the main molecular chain of NHAWP is polymerized by AM and AMPS. The AM can hydrolyze carboxyl groups in strong alkaline environment of the cement slurry. Meanwhile, AMPS has sulfonic acid groups, which can adsorb calcium ions in cement slurry. However, after the polymerization of AM and AMPS, only little amount of AM and AMPS remain in the cement slurry. Therefore, as expected, the NHAWP shows little effects on the thickening property of the cement slurry. Please see page 20, line 546-552.

Revisions have been made in the manuscript:

(see page 20, line 546-552)

Besides, after incorporating NHAWP, the thickening time of cement slurry is prolonged, but the change is not obvious. This is due to the main molecular chain of NHAWP is polymerized by AM and AMPS. The AM can hydrolyze carboxyl groups in strong alkaline environment of the cement slurry. Meanwhile, AMPS has sulfonic acid groups, which can adsorb calcium ions in cement slurry. However, after the polymerization of AM and AMPS, only little amount of AM and AMPS remain in the cement slurry. Therefore, as expected, the NHAWP shows little effects on the thickening property of the cement slurry.

Q6. Abstract section. The abbreviation KWHL-1 is neither appropriate nor definite.

Authors' reply: Many thanks for your comments. The abbreviation KWHL-1 has been revised to NHAWP, which represents new kind of hydrophobically associating water-soluble polymer. This has been revised throughout the manuscript. Please see page 1, line 12-26.

Revisions have been made in the manuscript:

(see page 1, line 12-26)

During the process of well cementing in deepwater, the cement slurry experiences a wide range of temperature variation from low temperature at seabed to high temperature in downhole. The elevated temperature affects the rheology of cement slurry. The change of rheology of cement slurry could influence the safety of cementing operation. The aim of this paper is to develop a new

kind of hydrophobically associating water-soluble polymer (NHAWP) as an additive to prepare a constant rheology oil well cement slurry, which can be used at temperature range from 4°C to 90°C. The acrylamide, 2-acrylamide-2-methylpropionic acid and stearyl methylacrylate were applied to synthesize the NHAWP by the inverse microemulsion polymerization. Test results indicate that the critical association temperature of NHAWP is 45°C. The critical association temperature is independent with NHAWP concentration, salt concentration and alkalinity of solution. When the temperature is below 45°C, NHAWP shows little influence on the viscosity of solution. When the temperature is above 45°C, the NHAWP forms spatial network structure by intermolecular hydrophobic association, and thus increases the viscosity of solution significantly. The NHAWP also displays good thermal stability and excellent salt and alkali resistance properties. In addition, the NHAWP shows nearly no negative influence on the basic properties of cement slurry, which indicates that the NHAWP can be used as a constant rheology agent to prepare a cement slurry with constant rheology in the temperature range of 4 to 90°C.

Q7. Can they definite KWHL-1 which corresponds to synthesize polymer reported in this work?

Authors' reply: Many thanks for your comments. The FTIR and NMR results can both definite the KWHL-1 (NHWP) which corresponds to the synthesize polymer. As shown in Fig. 5, the absorption peaks at 3342 cm^{-1} and 3195 cm^{-1} are the stretching vibration peaks of N-H in AM. The peak at 2921 cm^{-1} is attributed to the antisymmetric stretching vibration of methylene. The peak at 2856 cm^{-1} is due to the symmetry stretching vibration of methylene. The stretching vibration of C=O in SMA can be observed at 1731 cm^{-1} . The stretching vibration of C=O in Acrylamide and AMPS can be observed at 1654 cm^{-1} . The peak at 1612 cm^{-1} is the bending vibration of N-H in Acrylamide and AMPS. The deformation vibration of methylene can be observed at 1450 cm^{-1} . The absorption peaks at 1188 cm^{-1} and 1039 cm^{-1} are attributed to sulfonic group. The peak at 1110 cm^{-1} is attributed to C-O-C in Er-20. However, the characteristic absorption peaks of C=C cannot be found in Fig. 5. Therefore, it can be concluded that NHWP was the product of the copolymerization of AM, AMPS, SMA and Er-20. Fig. 6 shows the NMR spectroscopy of NHWP. As we can see, the chemical shift of peak a is at 3.61 and 3.67 ppm, which represents the chemical shift of H on $\text{CH}_3\text{-C-CH}_3$ in the AMPS structure. The H on CH_2 in the polymer backbone shows a chemical shift at 2.17 ppm (peak b). For peak c at 1.6 ppm, it is the chemical shift of H on CH_2 in the long carbon chain of the no-ionic emulsifier ER-20. The peak d at 1.45 ppm is from the H on CH_2 in the C-O-C chain of ER-20. Where the peak at 1.15 and 1.03 ppm are from the H on CH_2 and CH_3 in hydrophobic monomer respectively. Moreover, the peak value of C=C cannot be identified in Fig. 6 (normally at 4.59 and 6.15 ppm), which illustrates that the polymer reaction is complete. This result well agrees with the FTIR spectra analysis.

Fig. 5 Infrared spectrum of NHAWP.

Fig. 6 NMR spectroscopy

Please see page 12-14, line 313-344.

Revisions have been made in the manuscript:

(see page 12-14, line 313-344)

To probe the molecular structure of NHAWP, FTIR spectra analysis were applied on the NHAWP sample, as shown in Fig. 5. The absorption peaks at 3342 cm⁻¹ and 3195 cm⁻¹ are the stretching

vibration peaks of N-H in AM. The peak at 2921 cm⁻¹ is attributed to the antisymmetric stretching vibration of methylene. The peak at 2856 cm⁻¹ is due to the symmetry stretching vibration of methylene. The stretching vibration of C=O in SMA can be observed at 1731 cm⁻¹. The stretching vibration of C=O in Acrylamide and AMPS can be observed at 1654 cm⁻¹. The peak at 1612 cm⁻¹ is the bending vibration of N-H in Acrylamide and AMPS. The deformation vibration of methylene can be observed at 1450 cm⁻¹. The absorption peaks at 1188 cm⁻¹ and 1039 cm⁻¹ are attributed to sulfonic group. The peak at 1110 cm⁻¹ is attributed to C-O-C in Er-20 [27]. However, the characteristic absorption peaks of C=C cannot be found in Fig. 5. Therefore, it can be concluded that NHAWP was the product of the copolymerization of AM, AMPS, SMA and Er-20.

Fig. 6 shows the NMR spectroscopy of NHAWP. As we can see, the chemical shift of peak a is at 3.61 and 3.67 ppm, which represents the chemical shift of H on CH₃-C-CH₃ in the AMPS structure. The H on CH₂ in the polymer backbone shows a chemical shift at 2.17 ppm (peak b). For peak c at 1.6 ppm, it is the chemical shift of H on CH₂ in the long carbon chain of the no-ionic emulsifier ER-20. The peak d at 1.45 ppm is from the H on CH₂ in the C-O-C chain of ER-20. Where the peak at 1.15 and 1.03 ppm are from the H on CH₂ and CH₃ in hydrophobic monomer respectively. Moreover, the peak value of C=C cannot be identified in Fig. 5 (normally at 4.59 and 6.15 ppm), which illustrates that the polymer reaction is complete. This result well agrees with the FTIR spectra analysis.

References:

[27] Y. Bu, H. Liu, A. Nazari, Y. He, W. Song, Amphoteric ion polymer as fluid loss additive for phosphoaluminate cement in the presence of sodium hexametaphosphate, J. Nat. Gas. Sci. Eng., 31 (2016) 474-480.

Q8. *Both Françoise Candau and Enrique Jiménez-Regalado are reported a lot of investigations related to the HAWP. Can they consider in this work?*

Authors' reply: Many thanks for your comments. The studies of Françoise Candau and Enrique Jiménez-Regalado have been considered and discussed in section 1. Muñoz-López et al. [17] reported an associating multiblock copolymer electrolyte mediated by radical addition-fragmentation chain transfer technique. The influence of hydrophobic length was illustrated to play an essential role in the rheological properties of copolymers. Jiménez-Regalado et al. [18] investigated the phase behavior of hydrophobically modified polyacrylamides. Results show that viscosity of the polymer system is depressed due to a local segregation between the two copolymers.

Please see page 3, line 73-77.

Revisions have been made in the manuscript:

(see page 3, line 73-77)

Muñoz-López et al. [17] reported an associating multiblock copolymer electrolytes mediated by radical addition-fragmentation chain transfer technique. The influence of hydrophobic length was illustrated to play an essential role in the rheological properties of copolymers. Jiménez-Regalado et al. [18] investigated the phase behavior of hydrophobically modified polyacrylamides. Results show that viscosity of the polymer system is depressed due to a local segregation between the two

copolymers.

References:

- [17] C.N. Muñoz-López, S. Díaz-Silvestre, J.G. Telles-Padilla, C. Rivera-Vallejo, C. St Thomas, E. Jiménez-Regalado, Synthesis, characterization and rheological properties of multiblock associative copolymers by RAFT technique, *Polymer Bulletin*, 77 (2020) 2539-2555.
- [18] E. Jiménez-Regalado, J. Selb, F. Candau, Phase Behavior and Rheological Properties of Aqueous Solutions Containing Mixtures of Associating Polymers, *Macromolecules*, 33 (2000) 8720-8730.

Q9. *Can consider the emulsifier Er-20 as a monomer? See the section 2.2.3.1.*

Authors' reply: Many thanks for your comments. The emulsifier (Er-20) can also act as a hydrophobic monomer, which can polymerize at the same time as SMA due to it possesses the C=C bond. This has been clarified in section 2.2.1.

Please see page 6, line 155-156.

The results of FTIR and NMR also prove that the emulsifier Er-20 can act as a monomer. The peak at 1110 cm^{-1} in FTIR spectra is attributed to C-O-C in Er-20. In the NMR spectroscopy, the peak c at 1.6 ppm is the chemical shift of H on CH_2 in the long carbon chain of the no-ionic emulsifier ER-20. The peak d at 1.45 ppm is from the H on CH_2 in the C-O-C chain of ER-20.

Please see page 12, line 320-321 and page 13, line 334-337.

Revisions have been made in the manuscript:

(see page 6, line 155-156)

Moreover, the emulsifier (Er-20) can also act as a hydrophobic monomer, which can polymerize at the same time as SMA due to it possesses the C=C bond.

(see page 12, line 320-321)

The absorption peaks at 1188 cm^{-1} and 1039 cm^{-1} are attributed to sulfonic group. The peak at 1110 cm^{-1} is attributed to C-O-C in Er-20.

(see page 13, line 334-337)

For peak c at 1.6 ppm, it is the chemical shift of H on CH_2 in the long carbon chain of the no-ionic emulsifier ER-20. The peak d at 1.45 ppm is from the H on CH_2 in the C-O-C chain of ER-20.

Q10. *Thermogravimetric analysis was performed at a temperature range from 30 to 600 °C. The Fig 5 revealed a large amount of residue (ca. to 20 wt.%) after 600 °C. Can it attribute to a specific compound of KWHL-1?*

Authors' reply: Many thanks for your comments. This is because the broken of molecular chain of KWHL-1 is a continuous process from 371.6°C , and it does not decompose completely even at 600°C . This has been clarified in section 3.3.

Please see page 14, line 355-357.

Revisions have been made in the manuscript:

(see page 14, line 355-357)

Therefore, beyond 310°C, the NHAWP is stable, which shows an excellent high temperature resistance property. It shall be noted that a large amount of residue can still be identified after 600°C. This is because the broken of molecular chain of NHAWP is a continuous process from 371.6°C, and it does not decompose completely even at 600°C.

Q11. *Authors reported that the absorption peak of C=C could not be found. What kind of peak expected after polymerization process?*

Authors' reply: Many thanks for your comments. The peaks represent the function group in monomers are expected to be found after polymerization process. For example, the stretching vibration peaks of N-H in AM, the antisymmetric stretching vibration of methylene, the symmetry stretching vibration of methylene, the stretching vibration of C=O in SMA, the stretching vibration of C=O in Acrylamide and AMPS, and the bending vibration of N-H in Acrylamide and AMPS, etc. This has been discussed in section 3.2.

Please see page 12-13, line 315-327.

Revisions have been made in the manuscript:

(see page 12-13, line 315-327)

The absorption peaks at 3342 cm⁻¹ and 3195 cm⁻¹ are the stretching vibration peaks of N-H in AM. The peak at 2921 cm⁻¹ is attributed to the antisymmetric stretching vibration of methylene. The peak at 2856 cm⁻¹ is due to the symmetry stretching vibration of methylene. The stretching vibration of C=O in SMA can be observed at 1731 cm⁻¹. The stretching vibration of C=O in Acrylamide and AMPS can be observed at 1654 cm⁻¹. The peak at 1612 cm⁻¹ is the bending vibration of N-H in Acrylamide and AMPS. The deformation vibration of methylene can be observed at 1450 cm⁻¹. The absorption peaks at 1188 cm⁻¹ and 1039 cm⁻¹ are attributed to sulfonic group. The peak at 1110 cm⁻¹ is attributed to C-O-C in Er-20.

Q12. *The addition of salt led to a diminution of the viscosity of polymer solution. Recently, Cesar Muñoz-López demonstrated the influence of salt on rheological properties of polyelectrolytes (J of Applied Polym. Sci., 2021, 138, 43, 51270).*

Authors' reply: Many thanks for your comments. The literature has been mentioned in section 3.4.3.

Please see page 17, line 442-443.

Revisions have been made in the manuscript:

(see page 17, line 442-443)

The addition of salt led to a diminution of the viscosity of polymer solution. Muñoz-López et al. demonstrated the influence of salt on rheological properties of polyelectrolytes [29]. To investigate the effect of salt environment on the viscosity-temperature relationship, the viscosity test of aqueous solution with 0.6% concentration of NHAWP was performed under different salt concentration conditions.

References:

[29] C. Muñoz-López, C. St Thomas, L.A. García-Cerda, C. Rivera-Vallejo, E. Jiménez-Regalado,

Impact of additives on the rheological properties of associating water-soluble multiblock polyelectrolytes, 138 (2021) 51270.

Q13. The process reported in this work is it reproducible?

Authors' reply: Many thanks for your comments. All the process reported in this work is reproducible. All the tests to evaluate the property of cement slurry were performed for three times, and the average value was calculated and recorded. This has been clarified in section 2.2.6.1.

Please see page 9, line 251-253.

The original experimental data of the rheological test have been provided in section 3.7.4 for reference.

Please see page 22, line 607-608.

Revisions have been made in the manuscript:

(see page 9, line 251-253)

The retarder, NHAWP and dispersant were mixed with cement. The defoamer and fluid loss additive were added into water. Cement slurry was prepared according to API 10B-2-2013. After that, all of the tests to evaluate the property of cement slurry were performed for three times, and the average value was calculated and recorded.

(see page 22, line 607-608)

The rheological properties of the basic cement slurry and cement slurry with NHAWP were studied at different temperatures. The results are shown in Fig. 15. The original viscometer parameters for calculating the rheological curve are provided in Table 3 and Table 4.

Table 3 Viscometer parameters for basic cement slurry

Temperature/°C	Φ_{300}	Φ_{200}	Φ_{100}
4	173	120	65
25	173	118	63
45	175	120	62
60	149	106	55
75	133	88	47
90	120	82	43

Table 4 Viscometer parameters for cement slurry with NHAWP

Temperature/°C	Φ_{300}	Φ_{200}	Φ_{100}
4	225	155	80
25	224	152	78
45	225	154	78
60	222	151	77
75	223	153	79
90	226	155	81

Reviewer: 2

Recommendation: This topic is very interesting. As the author said, a constant rheology agent is very important for the cementing engineering of oil and natural gas wells. But this paper contain some unconvincing data, authors need provide more experimental procedure and original experimental data. In this work, authors used the monomer (SMA, AMPS, AM) to develop a new hydrophobically associating water-soluble polymer (KWHL-1), and they studied the effects of the synthetic polymer on the consistency, compressive strength, and rheological properties of cement slurry. From the results of the manuscript, the authors are proving that KWHL-1 is beneficial to improve the performance of cement slurry.

Q1. However, combining the results of consistency and rheological properties of the cement slurries, which confused me. According to the consistency results, the consistency of cement slurry with KWHL-1 increased with temperature rise, but the rheological properties of the cement slurry have no change with increasing temperature. The two results are inconsistent.

Authors' reply: Many thanks for your comments. As we can see in Fig. 13, the consistency of cement slurry with KWHL-1 (NHAWP) is basically around 20 Bc. The increase of temperature shows little effects on the consistency of cement slurry with KWHL-1 (NHAWP), which is consistent with the rheological properties.

Fig. 13 Thickening curve of basic cement slurry and cement slurry with NHAWP.

Q2. Additionally, the rheological properties of the cement slurry are too excellent to believe. Authors need provide more experimental procedure and original experimental data.

Authors' reply: Many thanks for your comments. More detailed information regarding the experimental procedure and original experimental data are provided in section 2.6.2.5 and section 3.7.4.

Please see page 11, line 297-300 and page 22, line 607-608.

Revisions have been made in the manuscript:

(see page 11, line 297-300)

Therefore, the shear force value of the cement slurry at a certain shear rate can be indirectly characterized by measuring the rotation angle of the inner cylinder. After that, the shear rate

(γ) and shear stress (τ) can be calculated by Eq. 1 and Eq. 2.

$$\gamma = 1.705n_r \quad (1)$$

$$\tau = 0.5099F\theta \quad (2)$$

Where n_r is the rotational speed of the viscometer, (unit: r/min); F is the spring constant; θ is the reading value of the viscometer.

(see page 22, line 607-608)

The rheological properties of the basic cement slurry and cement slurry with NHAWP were studied at different temperatures. The results are shown in Fig. 15. The original viscometer parameters for calculating the rheological curve are provided in Table 3 and Table 4.

Table 3 Viscometer parameters for basic cement slurry

Temperature/°C	Φ_{300}	Φ_{200}	Φ_{100}
4	173	120	65
25	173	118	63
45	175	120	62
60	149	106	55
75	133	88	47
90	120	82	43

Table 4 Viscometer parameters for cement slurry with NHAWP

Temperature/°C	Φ_{300}	Φ_{200}	Φ_{100}
4	225	155	80
25	224	152	78
45	225	154	78
60	222	151	77
75	223	153	79
90	226	155	81